# Regulation of Milk Fat Synthesis: Key Genes and Microbial Functions

**DOI:** 10.3390/microorganisms12112302

**Published:** 2024-11-13

**Authors:** Ye Yu, Runqi Fu, Chunjia Jin, Huan Gao, Lin Han, Binlong Fu, Min Qi, Qian Li, Zhuo Suo, Jing Leng

**Affiliations:** 1Faculty of Animal Science and Technology, Yunnan Agricultural University, Kunming 650201, China; yy09091823@163.com (Y.Y.); fandrunqi@163.com (R.F.); jin_chunjia@ynau.edu.cn (C.J.); gaohuanhhhh@163.com (H.G.); 19286920591@163.com (L.H.); binlongfu@126.com (B.F.); qimin538@gmail.com (M.Q.); 13408830996@163.com (Q.L.); sz1509682@163.com (Z.S.); 2Key Laboratory of Animal Nutrition and Feed Science of Yunnan Province, Yunnan Agricultural University, Kunming 650201, China

**Keywords:** milk fat, genes, microorganisms, metabolites, cows

## Abstract

Milk is rich in a variety of essential nutrients, including fats, proteins, and trace elements that are important for human health. In particular, milk fat has an alleviating effect on diseases such as heart disease and diabetes. Fatty acids, the basic units of milk fat, play an important role in many biological reactions in the body, including the involvement of glycerophospholipids and sphingolipids in the formation of cell membranes. However, milk fat synthesis is a complex biological process involving multiple organs and tissues, and how to improve milk fat of dairy cows has been a hot research issue in the industry. There exists a close relationship between milk fat synthesis, genes, and microbial functions, as a result of the organic integration between the different tissues of the cow’s organism and the external environment. This review paper aims (1) to highlight the synthesis and regulation of milk fat by the first and second genomes (gastrointestinal microbial genome) and (2) to discuss the effects of ruminal microorganisms and host metabolites on milk fat synthesis. Through exploring the interactions between the first and second genomes, and discovering the relationship between microbial and host metabolite in the milk fat synthesis pathway, it may become a new direction for future research on the mechanism of milk fat synthesis in dairy cows.

## 1. Introduction

Milk provides humans with energy, high-quality protein, and the main minerals and is known as “the food closest to perfection” and “white blood”. Nowadays, in pursuit of a healthy life, the nutritional elements in milk, such as milk protein and milk fat, have become a source of wide concern, and the safe and efficient production of milk has become a hot topic for consumers worldwide. Milk fat is one of the very important components, which is mainly composed of triglycerides, phospholipids, sterols, and ketones. Milk fat synthesis and fatty acid composition may be influenced by a variety of factors such as genetics and metabolites [1]. There is evidence that milk fat can reduce the risk of central core obesity [2]. Some medium and short-chain saturated fatty acids from milk fat, such as C4:0, C8:0, C10:0, and C12:0, have been found to have anticancer, antiviral, and antimicrobial properties, as well as the ability to retard tumor growth [3]. Oleic acid is the most abundant unsaturated fatty acid in milk and can lower plasma levels of cholesterol, low-density lipoprotein cholesterol (LDL-c), and triacylglycerol (TG) [4]. Similarly, conjugated linoleic acid and C18:2 have antioxidant effects on hydrogen peroxide-induced oxidative cellular damage [5]. These positive effects reveal that the fatty acid composition of milk is one of the important factors affecting human health. It is of great significance to improve the composition and proportion of beneficial fatty acids through modern molecular nutritional techniques to further enhance the nutritional value of milk and to elucidate the anabolic mechanism of milk fat [6].

The milk quality of dairy cows is affected by a variety of factors (Figure 1), and how to improve the milk fat of dairy cows has always been a hot topic of research in the industry. Milk fat synthesis is a complex biological process involving multiple organs and tissues (Figure 2). In recent years, it has been found that ruminal microorganisms, which are both symbiotic and heritable in individual animals, can help regulate milk fat production and quality by producing nutrients in dairy cows that can only be found in difficult-to-digest food sources [7,8]. The present article reviews the role of genes, ruminal microorganisms, and metabolites in regulating milk fat secretion in dairy cows, with a view to laying the foundation for research related to improving milk fat in dairy cows.

## 2. Interaction of Two Genomes with Milk Fat Synthesis

Ruminants have an extremely complex ruminal microecosystem; the synthesis of milk fat is the result of the coordinated efforts of the gastrointestinal microorganisms and the cow’s organism (Figure 3). However, most of the current studies have focused on the functional genomes related to the synthesis of milk components (the first genome), whereas the studies related to the gastrointestinal microbial genome (the second genome) have only just begun, and the interactions between the two genomes have not yet been systematically investigated for their influence on the synthesis of milk components.

### 2.1. Key Genes Involved in Milk Fat Synthesis in the First Genome

The relationship between lactation-related genes and milk quality and their regulatory mechanisms is an important basic scientific issue. With the continuous development of molecular biology technology, the research on lactation biology of dairy cows has also shown a rapid development trend, and many lactation-related functional genes have been discovered. Since the completion of whole-genome sequencing of cows in 2006, researchers have carried out studies on lactation biology at the levels of lactation-related transcriptomics, the genome, and other bioinformatics and achieved many important results, which have revealed the correlation between polymorphisms of genes related to the synthesis of milk fatty acids and the milk fat rate, which has improved the milk quality and contributed to the healthy and rapid development of the dairy industry. This part reviews the relevant research progress in recent years at home and abroad on the relationship between fatty acid synthesis in de novo, fatty acid uptake and transport, triglyceride synthesis, lipid droplet synthesis, and secretion-related genes and milk fat and lays the foundation for further research on the mechanism of milk fat synthesis in dairy cows and the excavation of more key genes for lactation (Figure 4).

#### 2.1.1. Key Genes Involved in the De Novo Synthesis of Milk Fat

In the pathway, acetate and β-hydroxybutyric acid (BHBA) are converted to acetyl coenzyme A and β-hydroxybutyric acid coenzyme A under the catalysis of acetyl coenzyme A synthetase (*ACSS*) in mammary cells, and then, short-chain fatty acids are synthesized ab initio by acetyl coenzyme A carboxylase (*ACACA*) and *FASN* for milk fat, and parts of medium-chain fatty acids are converted to monounsaturated fatty acids by stearoyl coenzyme A dehydrogenase (*SCD*), resulting in an increased volume of adipocytes [9]. The high expression of *ACACA* and FASN genes in the mammary gland increased milk fat content and altered milk fatty acid composition in dairy cows [10]. *ACACB* is a key regulator of fatty acid oxidation and correlates significantly with milk fat content during lactation [11,12]. Interestingly, *SCD* is a key enzyme involved in lipid metabolism, and bovine SCD is highly expressed in the mammary gland; individuals with the GG genotype had significantly higher milk fat and milk protein content than those with the AA genotype [13]. It is worth noting that the rate-limiting enzymes of fatty acid synthesis, *ACACA*, *FASN*, and *SCD*, are important transcriptional regulators [14,15].

#### 2.1.2. Fatty Acid Uptake and Transport

Blood is the main source of long-chain fatty acids and monounsaturated fatty acids. Fatty acids taken up by mammary cells from the blood are hydrolyzed by the very low-density lipoprotein receptor (*VLDLR*) and components of the coelom [16]. The degradation of triglycerides to small molecules of glycerol and fatty acids is catalyzed by fatty acid transport and uptake enzymes such as lipoprotein lipase (*LPL*), adenosine triphosphate-binding cassette transporter A1 (*ABCA1*), adenosine triphosphate-binding cassette transporter protein G2 (*ABCG2*), fatty acid-binding protein 3 (*FABP3*) and acyl coenzyme A-binding proteins (*ACBPs*). Among them, *FABP3* regulates the major synthetic channels of milk lipids, and LPL is involved in the process of milk lipid synthesis and metabolism [16,17]. Notably, *ACBPs* mainly act in the transport and storage of long-chain acyl coenzyme A esters, which play an important role in lactation, and studies have shown that the expression of *ACBPs* is interconnected with the milk lipid metabolic process during lactation [18].

#### 2.1.3. Triglyceride Synthesis

Milk fat is an important nutrient and one of the components of the unique flavor of cow’s milk. TG makes up the largest proportion of milk fat, accounting for about 98% of the total lipids [19]. TG in milk fat is synthesized in the endoplasmic reticulum, and nascent TG is encapsulated by the endoplasmic reticulum membrane to form intracellular milk fat droplets. There are three steps in the synthesis of triglycerides, catalyzed by glycerol-3-phosphate acetyltransferase (*GPAT*), phosphoglycerol acyltransferase (*AGPAT*), and dilipoylglycerol acyltransferase (*DGAT*). Fatty acids and the de novo synthesis of esteroyl coenzyme A are esterified by *GPAT*, *AGPAT*, and *DGAT*, respectively, to produce triglycerides, which are then formed into milk fat particles and secreted by mammary epithelial cells into the lumen of mammary follicles.

Among the four subunits of mammalian *GPAT*, *GPAT1* and *GPAT4* are mainly expressed in mammary tissues. *GPAT1* and *GPAT4* polymorphisms are significantly correlated with milk production traits such as milk fat content, short- and medium-chain saturated fatty acid content, and unsaturated fatty acid content [20], which makes the *GPAT* gene polymorphisms a valuable tool for improving the fatty acid profile of milk. *DGAT1* is an important enzyme in mammals, a key and rate-limiting enzyme that catalyzes the synthesis of triglycerides, whose main role is to generate triacylglycerol from diacylglycerol plus fatty acid acyls. Meanwhile, the components of its genotype AK had significantly higher milk fat content than genotype AA, which increased by 0.31% when base A was mutated to K [21]. There is a strong link between the fatty acid content of milk and that of *DGAT1* and *SCD* genes. DGAT1 catalyzes the final stage of glycerol triacyl synthesis [22], whereas *SCD*, by introducing a double bond at the delta-9 position of C14:0, C16:0, and C18:0, is mainly involved in the synthesis of monounsaturated fatty acids [23]. The *AGPAT6* gene is involved in TG synthesis in mammary epithelial cells (MECs) and affects the expression of major genes associated with FA transport and activation, TG synthesis and transcriptional regulation, FA oxidation, and TG degradation during milk lipogenesis [24,25]. Mutations in the promoter region of the long-chain acyl coenzyme A (*ACSL1*) gene can significantly increase milk fat content, and its overexpression significantly raises the triglyceride content of adipocytes [26].

#### 2.1.4. Secretion of Lipid Droplets

Xanthine dehydrogenase (*XDH*), periplasmic lipid droplet proteins 1/2/3 (*PLIN1*/*PLIN2*/*PLIN3*), and butyrophilin subfamily 1 member A1 (*BTN1A1*) are the key genes for lipid droplet formation, which positively regulate the formation of lipid droplets and promote the synthesis and secretion of milky lipids [27,28]. Previous studies identified *PLIN3* as a lipid droplet-binding protein, which is mainly enriched in the membrane of intracellular lipid droplets centered on TG [29]. *PLIN3* protein levels are positively correlated with TG levels in the cell, and the inhibition of *PLIN3* expression prevents the development of intracellular lipid droplets and reduces the admixture of TG to the droplets, thereby affecting milk fat content [30]. *BTN1A1* has a significant effect on the content of fat and protein in milk; the overexpression of *BTN1A1* promotes lipid droplet synthesis in mammary epithelial cells [31].

In addition, a part of genes plays an important role in the secretion of lipid droplets, such as the prolactin (PRL) gene family and pituitary transcription factor 1 (*POU1F1*). The amino acid codon A-G mutation at exon 3, position 103 of the PRL gene produces polymorphism, and this difference between buffaloes and cows may be one of the reasons for the low milk yield and high milk fat and milk protein content in buffaloes [32]. Among them, RPL23A is an important key gene affecting the milk fat and protein rate in dairy cows [12]. *POU1F1* positively regulates the expression of growth hormones and prolactin mainly in the anterior pituitary gland of mammals, which in turn affects milk production traits; polymorphisms of the enzyme cleavage locus (A→G) on exon+dominant 6 were significantly associated with milk fat content in Holstein cows, with individuals of the AG genotype having the highest milk fat content [33,34].

In summary, it is necessary to analyze signaling pathways, transcription factors, enzymes, and genes related to milk fat synthesis with the help of genomics, transcriptomics, epigenomics, proteomics, metabolomics, and lipidomics and to further study the regulatory mechanism of milk fat synthesis, which is of great scientific significance for the full utilization of the lactation potential of cows and the production of high-quality milk.

### 2.2. Key Genes Involved in Milk Component Synthesis in the Second Genome

#### 2.2.1. Key Microbial Enzyme Genes

With the development of biotechnology, more and more researchers have begun to study the dairy cow host genome and ruminal microbial genome from the perspective of co-operation and to explore the role of ruminal microbiota and their interactions in the involvement of milk fat synthesis, milk production, and lipid metabolism using macrogenomics, macrotranscriptomics, and macroproteomics. Some of the functional genes involve in the synthesis of milk components have been screened from the second genome (gastrointestinal microbial genome). Most plant-based feeds are high in cellulose, and their degradation in the rumen requires catalysis by enzymes secreted by cellulose-degrading microorganisms. These cellulose-degrading microorganisms are capable of degrading cellulose to produce volatile fatty acids (VFAs) [35], such as acetate, propionate, and butyrate, and an increase in the concentration of acetate and butyrate provides sufficient precursors for the synthesis of milk fat and an increase in milk fat production [15,36]. Cellulase is essentially a complex enzyme consisting of a variety of enzymes, mainly including three endo-β-1,4-glucanases, exo-β-1,4-glucanases, and β-glucosidases, which act synergistically to accomplish the degradation task [37,38,39,40]. Currently, cellulose and hemicellulose degrading enzymes have been screened from the rumen of a variety of ruminants [41,42,43,44].

Ruminal microbes can be regarded as a resource pool of novel multifunctional cellulase genes. Currently, it has been found that microorganisms capable of degrading cellulose and secretory cellulase mainly include that of *Fibrobacter succinogenes*, *Ruminococcus albus*, *Ruminococcus flavefaciens*, and *Butyrivibrio fibrisolvens*. Meanwhile, anaerobic fungi degrading cellulose include Neocallimastigomycota, Neospora, Cyclospora, and so on [45,46]. Plant cell wall degrading enzymes in the rumen of musk oxen were investigated, and plant-cell-wall-degrading enzyme modules, including glycoside hydrolases, carbohydrate esterases, and polysaccharide cleavage enzymes, were identified from more than 2500 sequence overlapping clusters via a metatranscriptomic approach [47]. Previous studies have identified bacteria and eukaryotes that degrade fiber in the rumen of dairy cows. Furthermore, there were 12,237 carbohydrate-activated enzymes (CAZymes) identified in the rumen, which consisted mainly of the GH family, and most of them were derived from *Prevotella*, *Ruminococcus*, and *Fibrobacter* [48]. Further studies showed that the glycohydrolase family (GHs)—GH5, GH6, GH7, GH8, GH9, GH10, GH12, GH44, GH45, GH51, GH61, and GH74—have cellulose hydrolysis activity [49]. Among them, GH7 family cellulases are mainly concentrated in the whole endocellulase lineage; cellulase species are more diverse in the GH5 family [50]; exoglucanases are abundant in the GH6 and GH7 families [51]; and β-glucosidases are mostly found in the GH3 family [52]. Under the joint action of ruminal microorganisms and their secreted cellulase, a large amount of acetate, butyrate, fatty acids, etc., are produced, which provide sufficient precursors for the synthesis of milk fat and an increase in milk fat production.

Recently, researchers have used techniques such as macrogenomics, macrotranscriptomics, and macroproteomics to study the diversity of ruminal fibrillase genes. The discovery of these genes opens the way for the discovery of more novel cellulases in the future. There is a large number of untapped cellulase genes in the rumen, and their isolation and expression in engineered bacteria would be very promising for improving milk fat synthesis.

#### 2.2.2. Ruminal Microbe–Host Interactions

Ruminants rely on ruminal microbial community fermentation to convert complex polysaccharides such as cellulose and hemicellulose, which form a major part of the plant, into usable nutrients [53,54,55]. The rumen is inhabited by hundreds of millions of microorganisms, mainly bacteria, archaea, fungi, and protozoa [56]. Bacteria are the most abundant, accounting for about 95% of the total microbial population; archaea are less abundant at 2–5%; and eukaryotic protozoa and fungi are less represented, totaling 0.1–1% [55]. The microbial community in the rumen is critical to the host’s metabolism and has a strong impact on its performance, physiology, and health, with ruminal microbes playing a key role in the formation of milk fat precursors. Changes in milk fat production and milk fatty acid composition are usually caused by alterations in ruminal fermentation and biohydrogenation pathways, and ruminal microbes play a crucial role in this process [57].

##### Functional Ruminal Microbiota

The ruminal microbiota plays an important role in the lactation performance of the host. Current studies have shown that the dominant phyla of bovine ruminal microorganisms are Bacteroidetes, Firmicutes, and Proteobacteria—of which Bacteroidetes mainly decompose non-fiber carbohydrates, while Firmicutes mainly decompose fibers—and that there is a strong positive correlation between the milk fat yield and the proportion of Bacteroidetes and Firmicute [58,59]. Another study has shown that the proportion of *Prevotella* in the Bacteroidetes and the milk fat yield were significantly negatively correlated [60]. The proportions of the Firmicutes and Bacteroidetes in the rumen of high-yielding cows were significantly higher compared with those of low-yielding cows [59]. Schwartzia is a genus in the Firmicutes, which is more abundant in cows with higher milk production [60]. It is more interesting to note that Bacteroidetes dominated the metabolic functions of first-lactation cows, and Firmicutes and Proteobacteria dominated the metabolic functions of second- and third-lactation cows [57]. Second- and third-lactation cows have more diverse ruminal microbial populations and enzymes than first-lactation cows [61]. The phylum Ascomycota also accounts for a large proportion of the ruminal microflora and plays an important role in carbohydrate digestion [62]. Studies have shown that the phylum Ascomycetes gradually becomes dominant in the rumen of ruminants as the proportion of concentrate in the ration increases [63,64,65,66].

Significant correlations were found between the relative abundance and bacterial diversity indices of some ruminal flora and lactation performance, and the correlations were mainly found in milk yield and milk fat content [8], and some studies have shown that ruminal bacterial abundance is negatively correlated with milk fat in dairy cows [67]. The relative abundance of ruminal *Prevotella* was significant when cows developed low milk syndrome, while the relative abundance of unclassified Lachnospiraceae, Veillonellaceae, and Pseudobutyri-vibrio significantly decreased, and both acetate and butyrate concentrations tended to decrease [68]. Milk fat content has also been correlated with the abundance of *Dialister*, *Megasphaera*, *Lachnospira*, and *Sharpea* in the rumen [69].

##### Acid-Producing Bacteria

The proliferation of acid-producing bacteria in the rumen may alter ruminal fermentation patterns and ultimately affect milk fat content. The relative abundance of Kiritimatiellaeota in the rumen of cows with high saturated fatty acid content was significantly higher than that of cows with low saturated fatty acid content, suggesting that Kiritimatiellaeota could be involved in milk fat metabolism in dairy cows [70]. In addition, *Vibrio butyrate* and *Vibrio pseudobutyrate* are important degraders of polysaccharides in the rumen, fermenting structural carbohydrates (hemicellulose, xylan, and pectin) to produce formate, butyrate, and acetate [71], and *Vibrio pseudobutyrate* populations were positively correlated with milk fat and milk yield [72]. A high abundance of starch-degrading bacteria promotes carbohydrate metabolism and increases VFA production [73]. It was also shown that *Prevotella* was negatively correlated with milk fat content, Fibrobacter was positively correlated with milk fat content, and Ruminalia was positively correlated with milk fat and milk protein concentration, which may be attributed to the presence of starch-degrading bacteria [58]. It was also found that the Eubacterium could utilize hemicellulose in the ration, which could increase the concentration of acetic and butyrate in the rumen to promote the synthesis of milk fat [74].

Most of the anaerobic fungi belong to the Neocallimastix frontalis [75]. Although they are in low levels in the rumen, which play an important role in the degradation of structural carbohydrates in feed plants due to their polysaccharide-degrading enzymes and thus play an important role in the degradation of structural carbohydrates in feed plants [76]. Anaerobic fungi on the one hand contribute to the loosening of cellulose in the rumen through the penetrating action of the pseudoroot system; on the other hand, grazing secrete cellulases to break down the cellulose into VFAs, and finally, the VFAs are directly absorbed by the animal.

##### Adaptation of Microorganisms to Periparturient Dietary Changes to Promote Milk Fat Synthesis

Periparturient dairy cows undergo a major shift in diets before and after calving, with forages dominating prepartum and concentrates dominating postpartum, and the change in diets leads to a change in ruminal microbiology, which in turn affects the lactation performance of dairy cows. The relative abundance of Bacteroidaceae in the rumen of dairy cows decreased significantly from prepartum to postpartum [77]. Meanwhile, the relative abundance of Prevotellaceae was significantly increased by higher protein and carbohydrate content in the postpartum diets of dairy cows [78,79,80]. The relative abundance of Firmicutes in the rumen of dairy cows also changed during the periparturient period, with the relative abundance of Christensenellaceae and Ruminococcus being higher in the prepartum period but significantly lower in the postpartum period [62,81]. Christensenaceae are associated with the fermentation of structural carbohydrates in the ration, with acetic and butyrates being the main end products [82]. The morphology of ruminal epithelial tissue in periparturient dairy cows may also change accordingly due to the structure of the ration and physiological status. It has been found that postpartum ruminal epithelium of dairy cows requires 6–8 weeks to adapt to changes in the postpartum ration and reach a proliferative steady state [83]. During this period, ruminal papillae develop rapidly and increase in size, increasing the absorption and transport capacity of the ruminal epithelium, and the ruminal papillae change the most during the last 2 weeks of the late periparturient period [84]. And the relative abundance of the phylum Ascomycetes including γ-Ascomycetes (Gammaproteobacteria) and Succinivibrionaceae in the rumen of dairy cows after calving increased significantly postpartum in response to changes in prepartum and postpartum rations.

The behavior and biological effects of a microbial community depend not only on its composition [85,86] but also on the biochemical processes and their interactions that occur within each microorganism [87,88], which are strongly influenced by the genome of each microorganism living in the community. Recent studies have found significant differences in ruminal microorganisms in dairy cows with different nutrients across the same genetic background, feeding management, and farming environment, suggesting that microorganisms play an important role in the conversion of nutrients from feed to milk nutrients by the animal in vivo, mainly in the synthesis of milk fat and milk protein [89]. An in-depth excavation of gastrointestinal microorganisms affecting milk fat content and quality in dairy cows and investigation of their regulatory mechanisms on host epigenetics can lay a theoretical foundation for modern genetic breeding selection in dairy cows.

##### Involvement of the Microbiota–Gut–Brain Axis in Host Metabolism

Gastrointestinal microorganisms coexist and co-evolve with the host and constitute a complex microecosystem with the gastrointestinal tract. Notably, the gut–brain axis of the microbiota is involved in important physiological activities such as feed degradation, nutrient absorption, and immune regulation of the host, which has a significant impact on the production traits and health of the host. Communication from the gut microbiome to the central nervous system (CNS) occurs primarily through microbial-derived intermediates, mainly acetate, propionate, and butyrate of VFAs, which have been shown to be involved in the regulation of host metabolism and gut motility via the blood circulation to brain tissue [90,91,92]. VFAs have a mediating role in microbiota–gut–brain axis crosstalk [93]. VFAs are intestinal metabolites with neuroactive properties that can affect the gut–brain axis and modulate gut- and brain-related functions through a variety of potential pathways [94,95].

The intestinal tract is an important place for digestion and absorption in animals, as well as the largest immune organ of the organism, playing an important role in maintaining the normal immune defense function of the organism. Previous trials have confirmed that VFAs play an important role in improving gastrointestinal health [96,97,98]. Interestingly, free fatty acid receptor 2 (FFAR2) and free fatty acid receptor 3 (FFAR3), which are specific receptors for VFAs, are expressed in many organs, including the gut and brain [99]. Functional VFAs receptors FFAR2 and FFAR3 are present in the CNS and peripheral nervous system (PNS) [93]. VFAs activate G protein-coupled receptors (GPRs) and play a role in gut–brain communication [100].

Gastrointestinal microbiota and metabolites play an important role in milk fat synthesis in dairy cows, which may regulate the expression of milk fat synthesis-related genes in mammary tissues, thereby modulating milk fat production. However, little is known about the mechanisms by which gastrointestinal microbes and their metabolites regulate milk lipid production through the entero-gastric, entero-hepatic, entero-brain, and epigenetic pathways, and thus, much research is needed.

## 3. Ruminal Metabolites

Alterations in the gastrointestinal microbiota can influence the composition of systemic metabolites [101]. An in-depth study of the metabolic mechanisms of ruminal microbes on carbohydrates, proteins, and sugars can be targeted to regulate ruminal microbial fermentation and promote the production of milk component precursors. For ruminal microorganisms, bacteria are major contributors to the production of volatile fatty acids and microbial proteins [102,103]. Metabolomics is widely used in the study of environmental impacts and their effects on dairy cows to elucidate the relationship between different biofluids [67]. There are interactions between ruminal microbiota and the mammary gland, where ruminal metabolites can enter the bloodstream through the intestinal epithelium into the mammary gland, which then influences the composition of milk secreted by the mammary gland [104]. Some typical metabolites are highly correlated with specific ruminal bacteria, suggesting a functional correlation between the milk microbiota and the associated metabolites.

### 3.1. Volatile Fatty Acids

The ruminal fermentation process can break down the nutrients in the ration into VFAs such as acetate, propionate, and butyrate, which provide about 70% of energy to ruminants, and acetate is one of the VFAs in the rumen [105]. Acetate not only serves as a precursor for milk fat synthesis and directly participates in ab initio fatty acid synthesis in the mammary glands but also acts as a signaling molecule to regulate mammary fatty acid metabolism, which then regulates the synthesis of milk fat [106,107]. Recently, studies have shown that the ratio of acetate to propionate is positively correlated with milk fat rate [58] and that the increase in the concentration of acetate and butyrate in the rumen provides sufficient precursors for milk fat synthesis and increased milk fat production [15].

Acetate is the main substrate produced in the rumen for synthesizing milk fat, and Ruminococcus ruminantium UCG-00I, Ruminococcus, and Ruminococcus ruminantium UCG-014 can be used to increase milk fat rates by producing higher levels of acetate [68]. Similarly, a previous study found that acetate was a major fermentation product of Acetobacter, unclassified_f Lachnospiraceae, Saccharofermentan, and Thermoactinomyces, and was one of the critical factors in the rumen contributing to the increase in the fat content of milk [108]. Previous studies have found that the addition of sodium acetate to dairy cattle rations increased milk fat production [107]. They further found that sodium acetate significantly increased milk fat production in dairy cows by increasing the production of ad libitum synthesized fatty acids (C ≤ 16) in the mammary glands, confirming the important role played by acetate in milk fat synthesis [109].

BHBA is one of the most important precursors for milk fat synthesis in ruminants and is involved in the de novo synthesis of fatty acids, which was found to be positively correlated with milk fat content and plasma BHBA concentration [110]. The addition of BHBA to in vitro cultured mammary epithelial cells of dairy cows increased the expression of ACACA, FASN, and SREBP1 and then increased the intracellular triglyceride content of the mammary cells and the formation of lipid droplets in a dose-dependent manner [111]. Research has also found that BHBA is an important precursor of fatty acids, which are involved in the ab initio synthesis of fatty acids, and increased SREBP1 mediated triglyceride secretion in bovine mammary epithelial cells [112].

### 3.2. Long-Chain Fatty Acids

The composition of milk lipids largely depends on the effects of ruminal metabolism, i.e., biohydrogenation, isomerization, hydrolysis, and cow metabolism (lipid mobilization, mammary uptake of plasma lipids, and de novo synthesis) [113,114]. Some ruminal microorganisms are able to alter milk fatty acid composition by changing unsaturated fatty acids to saturated fatty acids through biohydrogenation. In vitro tests point to Butyrivibrio, Clostridiales, and Ruminococcaceae butyrivibrio as the major biohydrogenating bacteria [115]. Meanwhile, Butyrivibrio in the rumen of buffaloes was positively correlated with average milk fat production [116]. High levels of C18:3x-3 obtained from forage are partially biohydrogenated to Cl8:1t11 during digestion and partially absorbed intact in the intestine and subsequently secreted into milk [117].

Genomic and metabolomic analyses of the intestinal tract revealed the association of the Firmicutes (Desulfocucumis, Anaerotignum, and Dolosiccus) with myristic acid, and the phylum Ascomycetes (Catenovulum, Comamonas, Rubrivivax, Marivita, and Succinimouas) was positively correlated with choline [108]. These interactions may be a major factor in the inhibition of methanogenic bacteria, producing less methane and thus increasing the efficiency of milk fat production. The difference in milk fat content may be caused by the interaction between gut microorganisms and their metabolites, especially Firmicutes-myristic acid and Proteus-cholin [108]. The search for alternative indicators of intestinal methane emissions has now been extended to compounds present in biological matrices, particularly fatty acids (FAs) in milk. The rationale for using milk FA is based on the known relationship that exists between the ruminal production of methane and the production of VFAs, particularly acetate and butyrate [118], which are precursors of milk FA in the mammary gland [119]. One possible reason for the lack of generalizability of predictive models is that milk FA is strongly influenced by diet [120], and just as importantly, the composition of the diet is known to influence methane emissions [121].

### 3.3. Ruminal Microorganisms and Their Metabolites Interact with Each Other

Alterations in the gastrointestinal microbiota affect metabolite production, as well as metabolic pathways in cows [122,123,124]. Therefore, changes in ruminal microbiota and their circulating metabolites may affect the overall physiological functions of the rumen and host systems, such as lactation performance [125]. The synergistic effect of ruminal microbiota and their metabolites in dairy cows may enhance the gene expression of hemicellulases, lipid synthases, and transferase enzymes, which in turn enhances the rumen’s ability to degrade nutrients from the ration and ultimately promotes an increase in milk fat content [8]. Eubacterium_xylanophium_group, Clostridium, and Ruminalococcus play an important role in the fermentation of cellulose-rich feeds and resistant starch and can produce acetate and butyrate to increase the production of volatile fatty acids, which in turn promotes the synthesis of milk fat [126,127]. It has been found that bacterial metabolites of Bacteroidetes (Bacteroidetes) can regulate the gluconeogenesis pathway by modulating bile acids, i.e., promoting cholesterol efflux through the dissociation of bile salts and thus regulating the concentration of triacylglycerol in the blood [128,129]. Lactobacillus, Ruminococcus, and Clostridium in the phylum Firmicutes have bile salt hydrolase activity, which produces acetate, lactate, and antimicrobial substances involved in the regulation of traits such as lactation organisms and immune responses [130,131].

Succinic acid in the rumen of dairy cows was positively correlated with *Prevotella*, and succinic acid can be involved in propionate metabolism [58]. Lauric acid was significantly positively correlated with *Coprcoccus*. Succinic acid was significantly positively correlated with Succinivibrionaceae and Prevotellaceae, and changes in these microorganisms can cause alterations in lipid and amino acid metabolic pathways in the rumen, which ultimately affects the milk fat content [68]. Phospholipids, amino acids, inorganic ions, dihydroxy acids, fatty acids, carbohydrates, cholesterol esters, and glycerides are the main metabolites in the rumen [8]. Under the same ration pattern, the different metabolites in the rumen of cows with different milk fat content were mainly lipids, organic acids, and the content of lauric acid, succinic acid, and ethyl laurate were higher in the high milk fat rate group than in cows with low milk fat content. Lauric acid promotes the synthesis of fatty acids in mammary epithelial cells, which in turn promotes the synthesis of milk fat [6]. Conjugated linoleic acid is an intermediate product of ruminal microbial metabolism, and the mechanism by which it can inhibit milk fat synthesis is related to the inhibition of gene expression of several enzymes encoding enzymes for single fatty acid uptake and transport, desaturase, and triglyceride synthase, which have the effect of decreasing milk fat synthesis.

Additionally, enrichment of the lipopolysaccharide biosynthesis pathway, amino acid biosynthesis pathway, valine biosynthesis pathway, leucine biosynthesis pathway, and isoleucine biosynthesis pathway significantly increased milk fat content [132]. Amino acid (including lysine, valine, leucine, and isoleucine) biosynthesis, lipopolysaccharide biosynthesis, cofactor and vitamin biosynthesis, phosphotransferase system, peptidoglycan biosynthesis, biotin metabolism, and amino sugar and nucleotide sugar metabolism were more highly enriched in buffaloes compared to cows [133].

## 4. Blood Metabolites

Metabolomics, which uses analytical methods based on untargeted or targeted approaches, may provide large-scale information on milk metabolites. Changes in the gastrointestinal microbiota are closely related to the systemic metabolic status of dairy cows [134]. Alterations in gastrointestinal microorganisms largely affect the metabolites present in the blood of the host animal [135]. Blood is a commonly used biofluid in metabolomics analyses for providing metabolites from all organs and linking them to metabolic diseases. Metabolites in serum or plasma are more representative of overall metabolic changes throughout the body of an organism. Metabolites that differed between groups in plasma samples from cows with different milk fat content were mainly enriched in the cholinergic synapse, glycerophospholipid metabolism, and Glycine, Serine, and Threonine metabolism, whereas intergroup differences in metabolites in milk with different milk fat content were mainly enriched in retrograde endocannabinoid signaling, starch and sucrose metabolism, and arginine and proline metabolism [136]. Blood metabolome analysis revealed that all carbohydrates and carbohydrate derivatives were enriched in the cows with high fat content, and in addition to citric acid, succinic acid was much higher in the serum of the high-dairy-fat cows, confirming that the tricarboxylic acid cycle is more active in the high-dairy-fat-content group and that pyruvic acid can be converted by oxidative decarboxylation to acetyl coenzyme A, which can be involved in the tricarboxylic acid cycle to produce adenosine triphosphate (ATP) to satisfy the higher energy demand of cows with high milk fat content, suggesting that cows with high milk fat content produce more energy for lactation through glucose oxidation [137]. It was found that plasma concentrations of BHBA and phosphatidylcholine were higher in cows with high milk fat content, and they could be taken up by the mammary gland for triglyceride synthesis; in addition, lower glycolytic metabolites within high milk fat content may be associated with milk fat metabolism compared to the content [138]. Plasma–milk intergroup metabolite analyses suggest that certain milk fat globule membrane components are metabolized in cows with different milk fat rates in an opposing trend and that phosphatidylcholine within samples of high milk fat content may be skewed toward the synthesis of glycerophospholipids, which, in turn, produces lower levels of sphingomyelin [139]. Blood metabolomics, as one of the effective ways of discovering and predicting economic traits in livestock, has been widely used in studies of multiple economic traits in several species [136,140,141]. However, the mechanism of discovering differences in milk production traits in dairy animals through serum or plasma metabolomics has been less studied.

## 5. Milk Metabolites

Milk is a complex biological fluid produced by the mammary gland, and its composition reflects the quality of milk and the metabolic processes and status of the mammary gland. Previous research has found that carnitine, choline, and citrate could be used as marker metabolites to differentiate Holstein cows from Jersey cows [142]. Meanwhile, further studies have investigated the milk metabolism of Holstein cows, Jersey cows, Yaks, goats, camels, and horses and found that the differential metabolites, choline and succinate, could differentiate dairy products from Holstein cows and other milk-producing animals [143]. These facts suggest that specific metabolites in milk can be used as markers to differentiate between different milks. Furthermore, an in-depth study investigated the metabolites and somatic cell count (SCC) of cow’s milk and found that the metabolites acetate, butyrate, isoleucine, and β-hydroxybutyric acid were increased and hippuric acid and fumaric acid were decreased in cow’s milk when the SCC of the cow’s milk was elevated and that these metabolites could be used as potential markers for the identification of the quality of cow’s milk [144]. Among triacylglycerol-rich lipids, milk contained higher levels of cortisol than lauric acid, and the above studies suggest that cortisol is associated with fat metabolism [145]. Another study measured the lactation performance of Holstein cows by feeding them forages of different qualities; only 61 metabolites were detected in the milk, including orotate, galactose, and linoleic acid, which are produced by the synthesis and degradation of lactation components [6]. Other studies have shown that microbial activity in the rumen affects the amount of AA in milk [146]. Both lower concentrations of α-aminoadipic acid and higher concentrations of spermidine were associated with higher milk protein content in Holstein cows [147]. In contrast, acetylcarnitine in milk was negatively correlated with methane emissions [148]. A correlation analysis of blood metabolites and milk metabolites revealed that nine identified metabolites in milk were also identified metabolites in plasma [149]. They were AA and its derivatives (leucine, isoleucine, valine, serine, tyrosine, and pyrrolidine), and they were higher in plasma. However, milk has higher levels of dimethyl sulfone, creatine, and citric acid. Dimethyl sulfone in milk fat also showed a correlation with plasma formic acid and hydrastine, two metabolites associated with ruminal microbes and methane [150].

## 6. Summary and Prospects

At present, some progress has been made in the mechanism and regulation of milk fat synthesis and secretion, but it is still incomplete. In particular, the mechanisms by which gastrointestinal microorganisms and host metabolites regulate milk fat production in dairy cows need to be further investigated. The maturation of histological analyses, systems biology, and epigenetic approaches will provide the basis for a deeper understanding of the mechanisms of milk fat production and milk fat synthesis in dairy cows. Exploring the interactions between the first and second genomes, microbial and host metabolites and milk fat in the milk fat synthesis pathway may become a new direction for future research on the mechanism of milk fat synthesis in ruminants.

## Figures and Tables

**Figure 1 microorganisms-12-02302-f001:**
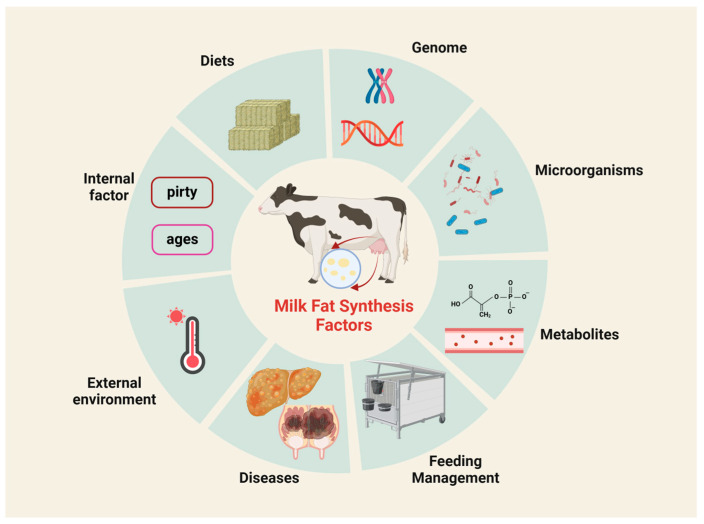
Factors affecting milk fat synthesis.

**Figure 2 microorganisms-12-02302-f002:**
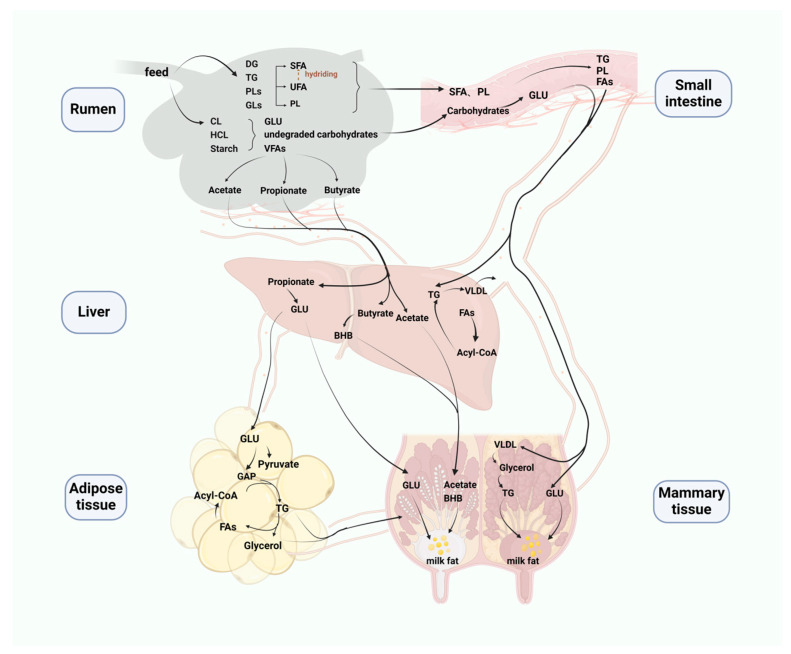
Schematic diagram of milk fat synthesis and secretion process. DG: diacylglycerol, TG: triacylglycerol, PLs: phospholipids, GLs: glycolipids, SFA: saturated fatty acid, UFA: unsaturated fatty acid, PL: phospholipid, FAs: fatty acids, CL: cellulose, HCL: hemicellulose, GLU: glucose, VFAs: volatile fatty acids, VLDL: volatile fatty acid, BHB: β-hydroxybutyric acid, GAP: glyceraldehyde-3-phosphate.

**Figure 3 microorganisms-12-02302-f003:**
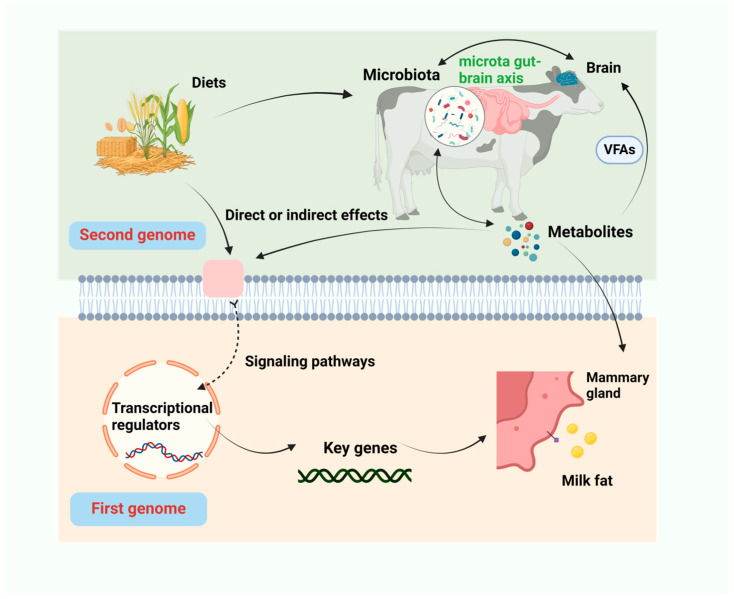
A sketch of the interaction of two genomes with milk fat synthesis.

**Figure 4 microorganisms-12-02302-f004:**
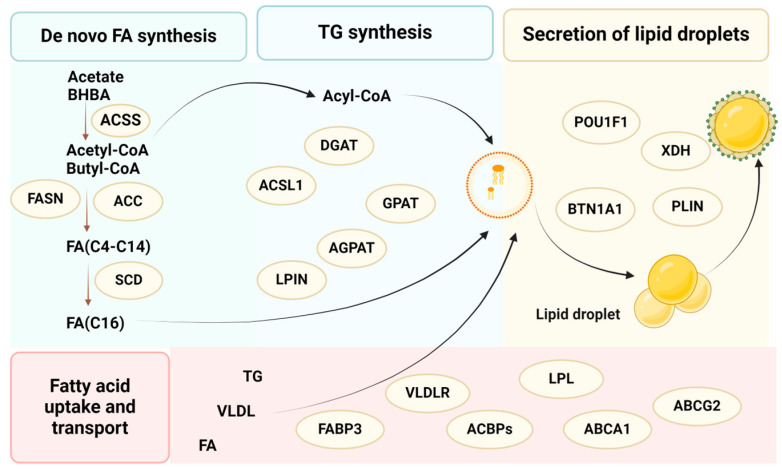
A sketch of key genes involved in milk fat synthesis.

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
