# Peer review of "Regulation of Milk Fat Synthesis: Key Genes and Microbial Functions"

_microorganisms, 2024, doi:10.3390/microorganisms12112302_

Round 1
Reviewer 1 Report
Comments and Suggestions for Authors
Manuscript microorganisms-3286928, entitled “Regulation of Milk Fat Synthesis: Key Genes and Microbial Functions”
Recommendation: The above paper is not suitable for publication in its present form.
This review article provides information on the regulation of milk fat synthesis. It is in general appropriately organized, carried out and written, however there are some points that should be corrected or clarified.
It will be very helpful to provide a figure with the key genes and/or microbial functions involved in the regulation of milk fat synthesis.
In several parts, authors should revise the sentences: L17-20, 96-98, 198-202, 284-287, 287-291, 336-338, 339-341, 414-416.
Please provide an explanation of abbreviations when they are initially used in text or a specific part where abbreviations are explained
In L27, please provide a title for the section (also numbering). Please check numbering in L230 (1.2.2.1.), 258, 281, 317, 345
Please check reference style of the journal
Specific comments
L12: “alleviating” instead of “mitigating”
L28: “high protein”? Do you mean high-quality protein?
L31: “consumers worldwide” instead of “global public”
L41: “These positive effects reveal” instead of “This shows”
L94: “expression” instead of “expressing”
L95: Altered compared to?
L111: Please add a reference
L131: “fatty acid profile” instead of “nutritional health”
L154: “…in milk, and over-expression of…”
L160-161: “…RPL23A is an important key gene affecting milk fat and protein rate…”
L172: I prefer “rumen microbiome” instead of “second”
L192: “…that of Bacillus…”
L217: Please delete. It is not related
L236: Please delete “Then”
L253: “…Prevotella was significantly increased when…”
L255: “…and Pseudobutyrivibrio significantly decreased, and both acetic…”
L260: “The relative…”
L275: “…they are in low levels in the rumen…”
L276: Please delete “library of”
L279: “secrete”
L320: “feed”
L361: “Which not only serves…” It refers to acetate?
L391-393: Repetition (third or forth time)
L457: Which purpose?
L494: “These facts suggested that…”
Comments on the Quality of English LanguageThe English could be improved to more clearly express the research
Author Response
1. Summary |
|
|
Thank you very much for taking the time to review this manuscript. On behalf of all the contributing authors, I would like to express our deepest respect and gratitude to the editor and reviewers for your hard work! As the reviewer was concerned, there are several issues that need to be addressed. We tried our best to improve the manuscript and made some changes to the manuscript, the detailed corrections are listed below and marked in red in the revised manuscript. We are very grateful to the reviewer for their enthusiastic work and hope that the correction will meet with approval.
|
||
2. Questions for General Evaluation |
Reviewer’s Evaluation |
Response and Revisions |
Is the work a significant contribution to the field? |
|
We feel great thanks for your professional review work on our article. |
Is the work well organized and comprehensively described? |
|
|
Is the work scientifically sound and not misleading? |
|
|
Are there appropriate and adequate references to related and previous work? |
|
|
Is the English used correct and readable?
|
|
|
3. Point-by-point response to Comments and Suggestions for Authors Comments 1: It will be very helpful to provide a figure with the key genes and/or microbial functions involved in the regulation of milk fat synthesis. Response 1: We think this is an excellent suggestion. We have added a figure with the key genes involved in the regulation of milk fat synthesis. Revised manuscript, line 84.
|
||
Comments 2: In several parts, authors should revise the sentences: L17-20, 96-98, 198-202, 284-287, 287-291, 336-338, 339-341, 414-416. |
||
Response 2: We have re-written this part according to the Reviewer’s suggestion. L17-20: There exists a close relationship between milk fat synthesis and genes, microbial functions, which is both the process of organic linkage between different tissues and organs of the cow organism and a result of organic integration between the cow organism and the external environment. Revised: There exists a close relationship between milk fat synthesis, genes and microbial functions, as a result of the organic integration between the different tissues of the cow's organism and the external environment. Revised manuscript, lines 17-19.
L96-98: ACACB gene were significantly correlated with the milk fat contents of the lactation, and in addition ACACB is a key regulator of fatty acid oxidation, ACACB and its SNPs may affect milk fat composition in milk. Revised: ACACB is a key regulator of fatty acid oxidation and correlates significantly with milk fat content during lactation. Revised manuscript, lines 93-95.
L198-202: Comtet-Marre et al. in a transcriptomics revealing bacterial and eukaryotic fibre degrading in the rumen of dairy cows bacterial communities in the rumen of dairy cows, they analysed and identified 12,237 carbohydrate-active enzymes (CAZymes) composed mainly of the GH family, most of which were derived from the Prevotella, Ruminococcus and Fibrobacter, confirming the key role of these genera in fibre degradation. Revised: Previous studies have identified bacteria and eukaryotes that degrade fiber in the rumen of dairy cows. Furthermore, there were 12,237 carbohydrate-activated enzymes (CAZymes) identified in the rumen, which consisted mainly of the GH family and most of them were derived from Prevotella, Ruminococcus, and Fibrobacter. Revised manuscript, lines 190-193.
L284-287: The relative abundance of Bacteroidaceae in the rumen of dairy cows decreased significantly from pre-partum to post-partum, which was mainly manifested as a decrease in the relative abundance of Bacteroidales and Bacteroidaceae. Revised: The relative abundance of Bacteroidaceae in the rumen of dairy cows decreased significantly from pre-partum to post-partum. Revised manuscript, lines 268-270.
L287-291: While the relative abundance of Prevotellaceae relative abundance increased significantly in the postpartum period, and the increase in the relative abundance of Prevotellaceae the digestion and utilization of relatively high levels of proteins and carbohydrates in the postpartum diets of dairy cows. Revised: The relative abundance of Prevotellaceae was significantly increased by higher protein and carbohydrate content in the postpartum diets of dairy cows. Revised manuscript, lines 270-271.
L336-338: VFAs affects gut-brain communication functions through interaction with G-protein coupled receptors (GPRs) interactions affect gut-brain communication functions. Revised: VFAs activate G protein-coupled receptors (GPRs) and play a role in gut-brain communication. Revised manuscript, lines 314-316.
L339-341: Gastrointestinal microbiota and metabolites play important roles in milk fat synthesis and other milk production traits in dairy cows, then regulate milk fat production in dairy cows by driving the expression of milk fat synthesis-related genes in mammary tissues through epigenetic mechanisms. Revised: Gastrointestinal microbiota and metabolites play an important role in milk fat synthesis in dairy cows, which may regulate the expression of milk fat synthesis-related genes in mammary tissues, thereby modulating milk fat production. Revised manuscript, lines 316-318.
L414-416: Alterations in the microbiota of the gastrointestinal tract affect metabolite shifts, alterations in the gastrointestinal tract microbiota in turn affect metabolic pathways in dairy cows. Revised: Alterations in the gastrointestinal microbiota affect metabolite production, as well as metabolic pathways in cows. Revised manuscript, lines 386-387.
Comments 3: Please provide an explanation of abbreviations when they are initially used in text or a specific part where abbreviations are explained. Response 3: We sincerely appreciate the valuable comments. We have examined the entire manuscript and have explained acronyms when they are first used in the text, such as VFAs (volatile fatty acids), MECs (mammary epithelial cells).
Comments 4: In L27, please provide a title for the section (also numbering). Please check numbering in L230 (1.2.2.1.), 258, 281, 317, 345 Response 4: As suggested by the reviewer, we have provided the title "1. Introduction" for this section. Then, we checked the numbering of the other headings and made changes.
Comments 5: Please check reference style of the journal Response 5: Thank you for pointing this out. Therefore, we have corrected the reference style of the manuscript.
Comments 6: L12: “alleviating” instead of “mitigating” Response 6: We think this is an excellent suggestion. We've replaced " alleviating " with " mitigating ". Revised manuscript, line 12.
Comments 7: L28: “high protein”? Do you mean high-quality protein? Response 7: Thanks to the reviewer's suggestion, which here refers to "high quality protein", we have modified it. Revised manuscript, line 27.
Comments 8: L31: “consumers worldwide” instead of “global public” Response 8: We sincerely appreciate the valuable comments. We've replaced “consumers worldwide” with “global public”. Revised manuscript, line 30.
Comments 9: L41: “These positive effects reveal” instead of “This shows” Response 9: We have changed "This shows" to " These positive effects reveal ". Revised manuscript, line 39.
Comments 10: L94: “expression” instead of “expressing” Response 10: We have changed " expressing " to "expression ". Revised manuscript, line 92.
Comments 11: L95: Altered compared to? Response 11: The high expression of ACACA and FASN genes in the mammary gland increased milk fat content and altered milk fatty acid composition in dairy cows.
Comments 12: L111: Please add a reference Response 12: We sincerely thank the reviewer for careful reading. We have added the reference. Revised manuscript, line 103.
Comments 13: L131: “fatty acid profile” instead of “nutritional health” Response 13: As suggested by the reviewer, we have corrected the “nutritional health” into “fatty acid profile”. Revised manuscript, line 126.
Comments 14: L154: “…in milk, and over-expression of…” Response 14: We have modified it as "…in milk, and over-expression of…”.
Comments 15: L160-161: “…RPL23A is an important key gene affecting milk fat and protein rate…” Response 15: We have amended it as "…RPL23A is an important key gene affecting milk fat and protein rate…”. Revised manuscript, lines 154-155.
Comments 16: L172: I prefer “rumen microbiome” instead of “second” Response 16: We sincerely appreciate the valuable comments. But after our consideration and perusal, we have retained the “second genome”.
Comments 17: L192: “…that of Bacillus…” Response 17: We sincerely thank the reviewer for careful reading. As suggested by the reviewer, we have revised the“that Bacillus”into“that of Fibrobacter succinogenes”.
Comments 18: L217: Please delete. It is not related Response 18: We have deleted “Ruminants have been domestic animals for more than 10,000 years”.
Comments 19: L236: Please delete “Then” Response 19: “Then” was deleted.
Comments 20: L253: “…Prevotella was significantly increased when…” Response 20: We sincerely appreciate the valuable comments. We have modified it as "…Prevotella was significantly increased when…”. Revised manuscript, line 237.
Comments 21: L255: “…and Pseudobutyrivibrio significantly decreased, and both acetic…” Response 21: We have corrected the “and Pseudobutyri-vibrio decreased significantly, and both acetic” into “and Pseudobutyri-vibrio significantly decreased, and both acetate”. Revised manuscript, lines 239-240.
Comments 22: L260: “The relative…” Response 22: We sincerely thank the reviewer for careful reading. As suggested by the reviewer, we have corrected the“There lative”into“The relative”. Revised manuscript, line 244.
Comments 23: L275: “…they are in low levels in the rumen…” Response 23: We have amended it as "... they are in low levels in the rumen…”. Revised manuscript, line 257.
Comments 24: L276: Please delete “library of” Response 24: “library of” was deleted.
Comments 25: L279: “secrete” Response 25: In our resubmitted manuscript, we have corrected the“secretes”into“secrete”. Revised manuscript, line 262.
Comments 26: L320: “feed” Response 26: We feel sorry for our carelessness. we have corrected the “food” into “feed”. Revised manuscript, line 299.
Comments 27: L361: “Which not only serves…” It refers to acetate? Response 27: Here "which" refers to acetate, which we have modified. Revised manuscript, line 336.
Comments 28: L391-393: Repetition (third or forth time) Response 28: We were really sorry for our careless mistakes. We have removed the duplicates.
Comments 29: L457: Which purpose? Response 29: Thanks for your careful checks. We are sorry for our carelessness. We have modified it to “Metabolomics, which uses analytical methods based on untargeted or targeted approaches, may provide large-scale information on milk metabolites”. Revised manuscript, lines 424-425.
Comments 30: L494: “These facts suggested that…” Response 30: We have amended it as "These facts suggested that…”. Revised manuscript, line 461.
|
||
4. Response to Comments on the Quality of English Language |
||
Response: Thanks for your suggestion. We have tried our best to polish the language in the revised manuscript. For example, lines 97-99, 456-460, 462-471 in the revised manuscript. |
||
We thank the reviewer again for professional review, constructive comments and valuable suggestions on our manuscript. |

Reviewer 2 Report
Comments and Suggestions for Authors
Thank you for the opportunity to participate in the review of the manuscript entitled "Regulation of Milk Fat Synthesis: Key Genes and Microbial Functions".
The manuscript tells about the factors and possibilities of milk fat synthesis in cows.
The manuscript has a typical review paper layout.The introduction adequately tells about the topic and is supported by the relevant literature. It is very important that the authors cite new publications from recent years. The manuscript is divided into several chapters, which makes it much easier to read and analyze the text. The description itself is very long and detailed, a bit boring at times, but it contains a lot of interesting and valuable information. A very important thing in this manuscript is the organization of the text.
After reading the manuscript, the reviewer states that the manuscript is well written and appropriate for Microorganisms. After taking into account minor comments from the reviewer and minor improvements to the manuscript, it can be forwarded to further publication stages.
Detailed comments:
The entire manuscript uses an incorrect way of citing literature in the text. This should be corrected.
Line 26. Please use keywords other than the phrases used in the manuscript title. This will increase the search capabilities of the article in the database.
Line 45. Citation needed.
Line 192, 263, 368… Names of genera and species of microorganisms should be written in italics.
Line 221. Unnecessary repetition of a sentence.
Line 246. that
Line 380, 393… in vitro should be written in italics
Author Response
1. Summary |
|
|
Thank you very much for taking the time to review this manuscript. On behalf of all the contributing authors, I would like to express our deepest respect and gratitude to the editor and reviewers for your hard work! As the reviewer was concerned, there are several issues that need to be addressed. We tried our best to improve the manuscript and made some changes to the manuscript, the detailed corrections are listed below and marked in red in the revised manuscript. We are very grateful to the reviewer for their enthusiastic work and hope that the correction will meet with approval.
|
||
2. Questions for General Evaluation |
Reviewer’s Evaluation |
Response and Revisions |
Is the work a significant contribution to the field? |
We feel great thanks for your professional review work on our article. |
|
Is the work well organized and comprehensively described? |
||
Is the work scientifically sound and not misleading? |
||
Are there appropriate and adequate references to related and previous work? |
||
Is the English used correct and readable?
|
||
3. Point-by-point response to Comments and Suggestions for Authors Comments 1: The entire manuscript uses an incorrect way of citing literature in the text. This should be corrected. Response 1: Thank you for pointing this out. Therefore, we have corrected the reference style of the manuscript.
|
Comments 2: Line 26. Please use keywords other than the phrases used in the manuscript title. This will increase the search capabilities of the article in the database.
Response 2: We have revised the key words to "milk fat; genes; microorganisms; metabolites; cows". Revised manuscript, line 25.
Comments 3: Line 45. Citation needed.
Response 3: We sincerely thank the reviewer for careful reading. We have added the reference. Revised manuscript, line 43.
Comments 4: Line 192, 263, 368… Names of genera and species of microorganisms should be written in italics.
Response 4: Thank you for pointing out this problem in manuscript. We have reviewed the entire manuscript and have changed the names of genera and species of microorganisms to italics.
Comments 5: Line 221. Unnecessary repetition of a sentence.
Response 5: We agree with this comment. We have removed the duplicates.
Comments 6: Line 246. That
Response 6: We are grateful to the reviewer for comments, we have corrected the mistake. Revised manuscript, line 232.
Comments 7: Line 380, 393… in vitro should be written in italics
Response 7: Thank you for pointing out this problem in manuscript, the word "in vitro" has been italicized in the text. Revised manuscript, lines 355, 366.
We thank the reviewer again for professional review, constructive comments and valuable suggestions on our manuscript.

Round 2
Reviewer 1 Report
Comments and Suggestions for Authors
Authors made the necessary amendments and I suggest the acceptance of their article